# Advancements in Hyperspectral Imaging and Computer-Aided Diagnostic Methods for the Enhanced Detection and Diagnosis of Head and Neck Cancer

**DOI:** 10.3390/biomedicines12102315

**Published:** 2024-10-11

**Authors:** I-Chen Wu, Yen-Chun Chen, Riya Karmakar, Arvind Mukundan, Gahiga Gabriel, Chih-Chiang Wang, Hsiang-Chen Wang

**Affiliations:** 1Division of Gastroenterology, Department of Internal Medicine, Kaohsiung Medical University Hospital, Kaohsiung Medical University, No. 100, Tzyou 1st Rd., Sanmin Dist., Kaohsiung City 80756, Taiwan; minicawu@gmail.com; 2Department of Medicine, Faculty of Medicine, College of Medicine, Kaohsiung Medical University, No. 100, Tzyou 1st Rd., Sanmin Dist., Kaohsiung City 80756, Taiwan; 3Department of Gastroenterology, Dalin Tzu Chi Hospital, Buddhist Tzu Chi Medical Foundation, No. 2, Minsheng Road, Dalin, Chiayi 62247, Taiwan; dm498778@tzuchi.com.tw; 4Department of Mechanical Engineering, National Chung Cheng University, 168, University Rd., Min Hsiung, Chiayi 62102, Taiwan; karmakarriya345@gmail.com (R.K.); d09420003@ccu.edu.tw (A.M.); 5Vel Tech Rangarajan Dr. Sagunthala R&D Institute of Science and Technology, No. 42, Avadi-Vel Tech Road Vel Nagar, Avadi, Chennai 600062, Tamil Nadu, India; vtu18599@veltech.edu.in; 6Department of Internal Medicine, Kaohsiung Armed Forces General Hospital, 2, Zhongzheng 1st Rd., Lingya District, Kaohsiung City 80284, Taiwan; 7School of Medicine, National Defense Medical Center, No. 161, Sec. 6, Minquan E. Rd., Neihu Dist., Taipei City 11490, Taiwan; 8Hitspectra Intelligent Technology Co., Ltd., 8F. 11-1, No. 25, Chenggong 2nd Rd., Qianzhen Dist., Kaohsiung City 80661, Taiwan

**Keywords:** hyperspectral imaging, head and neck cancer, meta-analysis, convolutional neural network, support vector machine, linear discriminant analysis, computer-aided analysis, machine learning

## Abstract

**Background/Objectives:** Head and neck cancer (HNC), predominantly squamous cell carcinoma (SCC), presents a significant global health burden. Conventional diagnostic approaches often face challenges in terms of achieving early detection and accurate diagnosis. This review examines recent advancements in hyperspectral imaging (HSI), integrated with computer-aided diagnostic (CAD) techniques, to enhance HNC detection and diagnosis. **Methods:** A systematic review of seven rigorously selected studies was performed. We focused on CAD algorithms, such as convolutional neural networks (CNNs), support vector machines (SVMs), and linear discriminant analysis (LDA). These are applicable to the hyperspectral imaging of HNC tissues. **Results:** The meta-analysis findings indicate that LDA surpasses other algorithms, achieving an accuracy of 92%, sensitivity of 91%, and specificity of 93%. CNNs exhibit moderate performance, with an accuracy of 82%, sensitivity of 77%, and specificity of 86%. SVMs demonstrate the lowest performance, with an accuracy of 76% and sensitivity of 48%, but maintain a high specificity level at 89%. Additionally, in vivo studies demonstrate superior performance when compared to ex vivo studies, reporting higher accuracy (81%), sensitivity (83%), and specificity (79%). **Conclusion:** Despite these promising findings, challenges persist, such as HSI’s sensitivity to external conditions, the need for high-resolution and high-speed imaging, and the lack of comprehensive spectral databases. Future research should emphasize dimensionality reduction techniques, the integration of multiple machine learning models, and the development of extensive spectral libraries to enhance HSI’s clinical utility in HNC diagnostics. This review underscores the transformative potential of HSI and CAD techniques in revolutionizing HNC diagnostics, facilitating more accurate and earlier detection, and improving patient outcomes.

## 1. Introduction

On a global scale, head and neck cancer (HNC) is the seventh most frequently occurring cancer [1], resulting in 325,000 deaths and more than 660,000 new cases yearly [2]. Squamous cell carcinoma (SCC), which originates from the epithelial lining of the oral cavity, throat, and larynx, accounts for about 90% of HNC cases [3]. Diseases of the upper aerodigestive tract, encompassing all lesions of the mucosal surfaces of the nasal cavity, oral cavity, and nasopharynx down to the larynx, hypopharynx, and trachea, are essentially considered to comprise cancers of the head and neck [4]. In 2030, 439,000 cases of oral and oropharyngeal cancer are expected to be reported, based on estimates from the World Health Organization (WHO) [5]. A retrospective study of 9950 patients looked at a number of risk variables for HNC and its subtypes [6]. It was discovered that the greatest independent risk factor for an elevated risk of HNC is smoking, and HNC also has synergistic correlations with alcohol consumption [6,7,8,9,10,11,12,13,14]. Human papillomavirus (HPV) infection is known to be a major risk factor for oropharyngeal cancers (OPCs), particularly in younger age groups [15]. Additional risk factors for HNC include dental plaque accumulation, poor dental hygiene, long-term irritation of the oral lining, dietary factors, a low body mass index, and UV light exposure. These are risks because they can alter toxin and carcinogen metabolism, which can lead to the development of HNSCC [16,17,18]. In 2008, Molina et al. revealed that the 5-year overall survival rate among HNSCC patients in the United States was 39.5%, with rates of 34% for men and 52% for women; comparatively, the 1-year overall survival rate among HNSCC patients was 84%, with rates of 59% for men and 62% for women [19]. From 2016 to 2018, there was a rise in the number of patients with advanced stages of HNC in the UK, with higher percentages of patients with advanced disease in Scotland, Wales, and Northern Ireland (65–69%) than in England (58%) [20]. When combined therapeutic approaches are used, the global 5-year survival rates for surgery alone, surgery + adjuvant therapy, and exclusive radiation therapy, with or without chemotherapy, range from 50% to 65% [21].

Artificial intelligence (AI) is becoming more widely used in diagnostics as a result of the collection of massive digital datasets from the scanning of pathological glass slides [22]. A large number of researchers are currently developing computer-aided diagnostic (CAD) models for cancer detection and diagnosis at an early stage [23,24,25,26,27,28,29,30,31,32,33]. These include machine learning (ML) algorithms trained on medical imaging data from CT, MRI, and PET scans to automatically detect worrisome lesions or areas suggestive of cancer. The increasing use of ML has generated considerable interest, backed by a growing body of evidence showing its wide applicability in various cancer types [34,35,36,37,38,39,40,41,42]. In a study by Rahman et al., the authors distinguished between normal and SCC tissue in oral histology images using a support vector machine (SVM); the resulting diagnostic accuracy was 100% [43]. Krishnan et al. used linear discriminant analysis (LDA) and neural networks to categorize oral premalignant lesions into normal and atrophy-causing oral submucous fibrosis cases, which resulted in diagnostic sensitivity and specificity values of 92.31% and 100%, respectively [44]. Dos santos et al. used a CNN to detect cancerous regions in samples of oral cavity tissues, resulting in an accuracy of 97.6% and precision of 91.1% [45]. Furthermore, various reports have demonstrated that machine learning models surpass traditional statistical methods when it comes to tasks associated with head and neck cancer [46,47,48,49,50,51]. Biosensors have also emerged as a promising tool in the diagnosis of cancers, using biological molecules to recognize and detect specific biomarkers associated with cancer. Wang et al. showed how to detect and quantify SCC-Ag levels using immobilized SCC-Ag antibodies and an interdigitated electrode sensor modified with titanium oxide (TiO_2_). The sensitivity of the TiO_2_-modified sensor was 1000 times higher than that of other substrates [52]. Using surface-enhanced Raman scattering (SERS), Vohra et al. were able to differentiate SCC from other cell lines with a sensitivity of 93% and specificity of 100% [53]. In 2018, Soares et al. detected human papillomavirus in HNC cases using a microfluidic-based geno-sensor and managed to differentiate between HPV16-positive and -negative HNC cell lines [54]. In 2020, Olivia et al. used HNC cell lines to detect MGMT gene methylation using a self-assembled monolayer matrix-based geno-sensor. With a limit of detection of 0.24 × 10^−12^ mol L^−1^ throughout a wide concentration range of 1.0 × 10^−11^ to 1.0 × 10^−6^ mol L^−1^, the geno-sensor showed a high degree of sensitivity [55]. With the use of SERS and the MD analysis of biomarkers, Edoardo et al. were able to identify HNC and uncover the underlying biomolecular process involved in disease-related marker adsorption on silver surfaces [56].

Although HNC detection with CAD systems may be beneficial, there are drawbacks to this technology as well. The efficacy of CADx hinges on proficient medical image analysis; such analysis is pivotal as it directly impacts the clinical diagnosis and treatment process [57]. Traditional red, green, and blue (RGB) imaging is limited to capturing the visible spectrum’s three diffuse Gaussian spectral bands (380–740 nm) [58]. According to Bueno et al., there are still limitations to the amounts of data that can be processed and the processing techniques that are applicable when whole-slide imaging (WSI) is used for image processing [59]. Drukker et al. observed that CAD systems’ performance does not always predict their impact when they are employed in radiologists’ clinical practice [60]. Furthermore, research by Chang et al. and Acharya et al. demonstrated that CAD systems are impractical for real-time applications and require complicated computer analysis [61,62]. Occasionally, CAD tools provide inaccurate information regarding images, making them inappropriate for the early identification of malignancy [63]. Neal et al. noted that initial implementations of artificial neural networks (ANNs) required considerable tuning to achieve satisfactory results [64]. Additionally, CAD models are ineffective when the user provides incorrect data inputs during data gathering [65,66,67]. However, hyperspectral imaging (HSI) emerges as a non-invasive technique capable of overcoming the challenges outlined above [68,69]. In HSI, it is possible to extract basic features like color and texture, as well as analyzing more complex semantic features [70].

HSI is a hybrid diagnostic modality that extracts spectroscopic data from images [71]. It combines digital imaging with spectroscopy to increase the spectral quality of images within and beyond the visible spectrum [72]. Goetz et al. reported that NASA initially developed HSI for use in Earth surveillance and space exploration [73]. When spatial and spectral information is acquired and each pixel’s 2D spectral data are identified, a hyperspectral image is created; as a result, the origin of each spectrum in the area of interest may be identified [74]. Further, without the need for scanning, three-dimensional hypercubes can be quickly captured using single-shot hyperspectral imagers [75]. Researchers have used a variety of HSI approaches, including filter-based, whisk-broom, push-broom, and snapshot sensors [76,77,78,79,80,81,82,83,84,85,86,87,88,89,90,91,92,93]. High-resolution spectral data can be obtained using push-broom hyperspectral sensors; however, the scanning acquisition design of these sensors makes it more difficult to produce geometrically correct mosaics from several hyperspectral swaths [94]. In generating spatial data with a pixel-oriented hyperspectral sensor, whisk-broom scanning is helpful; nevertheless, because of the longer measurement length, it poses problems when it is used in outdoor cultural heritage situations [95]. The filter-based approach uses optical filters to capture spectrum information by means of wavelength-coded imaging [96]. Wu et al. developed a novel HSI technique known as snapshot hyperspectral volumetric microscopy, which collects data in five dimensions for the analysis of biological components [97]. HSI has seen recent use in various sectors, including medical diagnostics [98,99,100,101,102], environmental monitoring [103,104,105,106,107], forestry [108,109,110,111,112], mining and geology [113,114,115,116,117], remote sensing [118,119,120,121,122], agriculture [123,124,125,126,127], counterfeit detection [128,129,130,131,132], the military [133,134,135,136,137], and astronomy [138,139,140,141,142]. HSI has been particularly useful in the field of medical diagnostics in the recognition and diagnosis of various kinds of cancer. The most recent studies regarding the application of HSI for cancer diagnosis are presented in Table 1.

Figure 1 presents an illustrative diagram of the key elements regarding HNC. This study examines recent research concerning HNC diagnosis and detection using HSI technology combined with CAD methods. It assesses how effective these combined methods are in detecting and diagnosing HNC, with an emphasis on their sensitivity, specificity, and accuracy. Additionally, it gives a brief overview of the research and makes suggestions based on a meta-analysis of several CAD methodologies.

## 2. Materials and Methods

This section describes the methods used to gather studies relevant to this review, especially those dealing with the use of HSI technology in the detection and diagnosis of HNC. It outlines the criteria for including and excluding studies to ensure the selection of relevant research.

### 2.1. Study Selection Criteria

The objective of this review was to analyze advancements in HNC diagnosis and detection using HSI techniques, concentrating on studies that fulfilled the following established criteria for inclusion: providing clear numerical outcomes, such as datasets, accuracy, sensitivity, and specificity; focusing on HNC detection using HSI; published between 2015 and 2024; published in journals with an H-index exceeding 75 and falling within either the first quartile (Q1) or second quartile (Q2); employing either prospective or retrospective methodologies; and written in English. Furthermore, this research excluded studies that met the following criteria for exclusion: lacking adequate information; and falling under any of the categories of stories, remarks, proceedings, study protocols, systematic reviews, meta-analyses, and conference contributions. QUADAS-2 was used as a method for evaluating the scientific merit of the papers under examination [156,157,158]. There are four sections to this method: “patient selection”, “index test”, “reference standard”, and “flow and timing”. Furthermore, an evaluation of “applicability” is included in the first three sections. There are three categories for each component: high, low, and uncertain risk of bias [159]. Different researchers have used the QUADAS-2 tool to check whether studies match with real-time application conditions [160,161,162,163]. Sounderajah et al. conducted a quality review of AI-centered DTA studies and found that many of the studies did not tackle the study topic, instead focusing on the application of AI algorithms to predict treatment success, metastasis, recurrence, or disease prognosis [164]. Adeoye et al. produced evaluation designs to validate non-invasive biomarkers for HNC detection using the QUADAS-2 tool, finding that numerous potentially valuable biomarkers for HNC had not been thoroughly assessed using the most precise techniques available [165]. Therefore, in this study, two authors (G.G. and A.M.) performed QUADAS-2 analyses on several papers related to HNC detection using HSI and found that 7 of the 1030 papers available complied with the inclusion and exclusion criteria (see Appendix A, Section 1, for further details on the inclusion and exclusion criteria).

### 2.2. QUADAS-2 Results

The results of the QUADAS-2 analyses of the seven studies examined in this review are presented in Table 2. Specific criteria, such as patient selection, the index test, the reference standard, flow and timing, and the possibility of bias, were used to evaluate each study. The supplemental study contains further information on the quality analysis, inclusion and exclusion criteria, and other related topics.

## 3. Results

This section presents the findings of the review, detailing the clinical features observed in each study and providing a concise explanation. It includes the numerical results obtained from each study and provides comparisons of these results in relation to sensitivity, specificity, and accuracy.

### 3.1. Clinical Features Observed in the Studies

The studies selected for this review analyzed different CAD methods for detecting and diagnosing HNC. Each study is briefly described herein, emphasizing its goals, the CAD algorithm utilized, and the outcomes. Furthermore, the accuracy, sensitivity, and specificity of HNC detection and classification in each study are presented using subgrouping.

In 2019, Halicek et al. used a CNN to detect the cancer margins of HNSCC. In total, 102 patients were included in this study. The accuracy rates in their findings varied depending on the anatomical site within the oral cavity, ranging from 61 ± 7% for the tongue to 95 ± 4% for the oropharynx. Variability was also observed in the sensitivity and specificity. For example, in the hypopharynx, the sensitivity was 20 ± 14%, while the specificity was an impressive 99 ± 1%. The imaging was performed in vivo, including a variety of locations such as the tongue, using a wavelength range of 450–900 nm and a resolution of 5 nm. Eggert et al. (2021) demonstrated the use of deep learning algorithms combined with HSI for the in vivo diagnosis of HNC. This study applied a 3D-CNN and was mostly concerned with oropharyngeal, hypopharyngeal, and laryngeal tumors, obtaining an overall accuracy of 81%, sensitivity of 83%, and specificity of 79%. The spectral range that was imaged was 390–680 nm, and the spectral resolution was 10 nm. In vivo data collection from 98 participants was performed for this investigation.

Halicek et al. (2017) used HSI to classify HNC. In this study, a CNN was trained on both SCCa and thyroid cancer. For SCCa, an accuracy of 74%, sensitivity of 67%, and specificity of 67% were obtained, while for thyroid cancer, an accuracy of 67%, sensitivity of 67%, and specificity of 67% were obtained. This study was conducted ex vivo with a wavelength range of 450–900 nm and resolution of 5 nm. Ma et al. (2022) used a CNN to detect HNSCC on the larynx, hypopharynx, buccal mucosa, and floor of the mouth, resulting in accuracy, sensitivity, and specificity values of 82%, 72%, and 93%, respectively. Twenty patients were examined in this study, and the experiments were conducted ex vivo in the spectral range of 470–720 nm, with a resolution of 3 nm. Halicek et al. (2020) detected malignancies in the salivary and thyroid glands using deep learning algorithms and HSI. Their CNN was able to identify salivary gland tumors with 82 ± 8% accuracy, 72 ± 11% sensitivity, and 82 ± 11% specificity. These tests were carried out ex vivo on 82 thyroid tissue samples and an undefined number of salivary glands. Petzborn et al. (2022) investigated the intraoperative evaluation of tumor margins in tissue slices using HSI and ML approaches. In this study, an SVM detected oral cancer with accuracy, sensitivity, and specificity values of 76%, 48%, and 89%, respectively, in the wavelength range of 500–1000 nm and with a resolution of 2 nm. This study included seven patients, and the experiment was conducted ex vivo.

In 2017, Fei et al. completed a pilot study using label-free reflectance HSI for tumor margin evaluations utilizing surgical tissues from cancer patients. This study encompassed the oral cavity and thyroid regions. By employing LDA, they achieved promising results. For the tumor margin assessment in the oral cavity, the LDA model exhibited 90%, 89%, and 91% accuracy, sensitivity, and specificity, respectively, within the wavelength range of 450–950 nm and with a 2 nm resolution. The LDA model performed even better in the thyroid region, with accuracy, sensitivity, and specificity values of 94%, 94%, and 95%, respectively. These tests were carried out in vivo on 16 surgical specimens.

Table 3 below illustrates the clinical characteristics of the selected studies on HNC detection and diagnosis. All seven journal manuscripts included in this review specified the number of patients involved in the study. Additionally, these studies offered full details on the methodology used to collect data from hyperspectral images. In total, 375 patients were involved in the investigations. The selected papers focused on wavelengths in the regions of visible and near-infrared light. The most commonly utilized CAD algorithm in the experiments was CNN, followed by SVM and LDA. In all the investigations, the highest accuracy (95%) was obtained for oropharyngeal cancer, while the lowest accuracy (42%) was obtained for hypopharyngeal cancer. With regards to the total number of patients across all the studies, Halicek et al. 2019 examined a large number of patients (27.2%), whereas Pertzborn et al. examined a small number (1.8%). Additionally, Fei et al. discovered greater levels of sensitivity and specificity for thyroid cancer than for other cancer types. Pertzborn et al. used a greater wavelength range than was used in previous research. Three experiments were conducted in vivo, while four were conducted ex vivo.

### 3.2. Meta-Analysis of the Studies

The average accuracy, sensitivity, and specificity values for the seven studies examined in this review were 77%, 68%, and 80%, respectively. Research conducted in vivo showed significantly greater accuracy (81%) and sensitivity (83%) but lower specificity (79%), as opposed to research conducted ex vivo, which demonstrated lower accuracy (79%) and sensitivity (65%) but higher specificity (91%). In total, 216 patients were involved in the in vivo studies, whereas 159 patients were involved in the ex vivo studies. Table 4 presents the averages obtained via a meta-analysis of all the studies and HNC subgroups.

ML and AI techniques have significantly improved healthcare, particularly in areas such as computer-aided diagnosis, retrieval, and analysis, and medical image processing [173]. In the meta-analysis, LDA was found to be the most effective CAD method for HNC diagnosis and detection, with 92% accuracy, 91% sensitivity, and 93% specificity. CNNs performed moderately, with 82% accuracy, 77% sensitivity, and 86% specificity, showing competence but slightly lower sensitivity when compared to LDA; despite this, CNNs were most commonly used in the selected studies. SVM had the lowest accuracy and sensitivity, 76% and 48%, respectively, but reasonably high specificity of 89%. Although LDA, CNNs, and SVMs are all AI approaches used to diagnose HNC, they have distinct properties. CNNs are used to process images and extract features [174]. LDA is able to project high-dimensional pattern samples to the ideal discriminant vector space, allowing it to extract classification information and reduce feature space dimensions [175]. SVMs are valued for their versatility and robustness in high-dimensional spaces [176]. LDA was employed in only one trial, whereas CNNs were used in five; nevertheless, the meta-analysis results demonstrated that LDA was superior to CNNs in HNC detection.

Every type of cancer has unique risk factors and causes, but there are certain anatomical similarities, as well as shared risk factors like alcoholism, smoking, and HPV infection [177,178,179,180,181,182,183]. In particular, oral, tongue, and laryngeal cancers share similar risk factors and anatomical proximity [184,185], while others like thyroid cancer diverge due to endocrine-related factors [186,187,188,189,190]. With an accuracy of 86%, sensitivity of 87%, and specificity of 84%, the detection of thyroid cancer exhibited the best overall performance characteristics. Conversely, hypopharynx cancer detection exhibited a notable specificity of 99%, despite its lower sensitivity of 20% and accuracy of 42%, which was the lowest accuracy observed among the cancer types. Max. sinus cancer detection exhibited the second-lowest accuracy of 58% but the highest sensitivity of 93% among the cancer types. The lowest specificity was found for max. sinus cancer (52%) and tongue cancer (53%).

Studies that used a spectral range in the visible (VIS) band exhibited better performance metrics, with accuracy, sensitivity, and specificity values of 81.5%, 77.5%, and 86%, respectively, as compared to those that used the visible and near-infrared (VNIR) band, which exhibited an average accuracy of 75.6%, sensitivity of 69.5%, and specificity of 76.07%. Even though the VIS band showed better results, it was used in only two studies, whereas the VNIR band was used in five.

The highest average accuracy of 86%, with a sensitivity of 87% and a specificity of 86%, was shown by two studies that were published between 2015 and 2017. Conversely, the three studies that were published between 2018 and 2021 demonstrated a slightly lower average accuracy of 81%, along with a sensitivity of 83% and a specificity of 79%. However, the two studies published between 2022 and 2024 showed a decrease in average accuracy to 79% and a substantial decrease in sensitivity to 60%, despite their high specificity of 91%. These findings show a potential temporal variance in diagnostic study outcomes, with earlier studies generating higher overall performance metrics than more recent ones, underscoring the necessity for ongoing research and improvement in head and neck cancer diagnosis.

### 3.3. Subgroup Meta-Analysis

The quantitative findings from this meta-analysis of HNC cancer research were organized graphically in a Deek’s funnel plot and forest plots. In Figure 2, forest plots display the sensitivity, specificity, and accuracy values for each CAD technique, experiment type (in/ex vivo), cancer type, spectral band, and publication year (see Appendix A). These analyses were performed at the 95% confidence level (Appendix A presents the ANOVA tables for the different subgroups). The average accuracy, specificity, and sensitivity values for the data used in each categorization were used to determine the line of no effect in this investigation. Low-performance data are defined as data that cross the line of no effect on the left-hand side. However, data that do not cross over the line of no effect are considered to perform well. The meta-analyses of the accuracy, sensitivity, and specificity showed that the experiment type (in/ex vivo), spectral band, method, and year of publication intersected with the line of no effect, indicating a high chance of comparable performance. In contrast, the cancer type subgroup showed deviations from the line of no effect, suggesting potential differences in diagnostic accuracy among anatomical regions.

A Deek’s funnel plot was created using a variety of classifications, including the CAD method, spectral band, experiment type (in/ex vivo), anatomical location, and year of publication (see Appendix A). The diagnostic odds ratio for each anatomical location and the fraction of the root of each sample size are displayed in the Deeks’ funnel plot in Figure 3 [191,192,193,194]. The Deek’s funnel plot obtained in this investigation did not indicate the presence of heterogeneity. The regression line for the anatomical location was found to be at approximately 12. The max. sinus, hypopharynx, and thyroid exhibited greater standard error values than other anatomical locations. Halicek et al. (2019) used HSI to detect SCC in different locations, including the oral cavity, larynx, nasal cavity, oropharynx, hypopharynx, and max. sinus, using the same dataset. A smaller dataset was found for salivary cancer compared to other cancer types in the investigation by Halicek et al. (2020).

A graph of the accuracy values in the examined research for the identification of the various HNC types was created to compare them graphically and is shown in Figure 4. This figure depicts the overall accuracy values for the various cancer types detected in the investigations. The most widely detected type was oral cancer. Oropharyngeal cancer was the most accurately detected among all the cancer types, with 95% accuracy, followed by thyroid cancer at 94%. In Halicek et al.’s (2019) study, hypopharyngeal carcinoma had the lowest accuracy of all, at 42%.

## 4. Discussion

HSI techniques can be applied as mesoscopic or microscopic imaging methods to monitor and analyze anatomical features on different scales, ranging from cells to tissues. Many studies have proven the capacity of HSI as a disease diagnostic tool in various tissues, including those of the head and neck [51,195,196,197,198,199,200]. Although preclinical and clinical research has shown the potential of HSI approaches, there are still significant challenges to overcome before HSI may be effectively used in clinical settings [201]. One of the challenges facing HSI is its sensitivity to external elements that might affect the quality and dependability of the collected data. These factors include atmospheric conditions, variations in lighting, and the distance between the imaging system and the object [202]. Various methods can be employed to mitigate the effects of external factors on HSI, particularly lighting conditions. These include standardized lighting setups, white and dark reference calibration before image acquisition, and post-processing algorithms to normalize spectral data. Reflectance-based approaches can also be used to reduce the impact of lighting inconsistencies. These techniques collectively enhance the reliability of HSI in different experimental and clinical environments, addressing its susceptibility to external influences. In addition, researchers need to address the challenges of obtaining high-resolution high-speed image collections at video rates when HSI is used in medical applications [203]. Target organ, tissue, cell, and molecular biomarker imaging during surgery would be made easier by real-time acquisition. Furthermore, with more spectral and spatial information, more subtle differences in the spectral and spatial characteristics of different tissue types may be captured by finer spectral and spatial resolution and a larger library of tissue spectra [204]. Wang et al. created a unique hybrid imaging system that takes advantage of the panchromatic camera’s high light throughput, along with the excellent spectral and spatial resolution of the CASSI, to capture 4D high-speed films [205]. The exclusion of the important NIR range (1000–1900 nm), which contains valuable information on lipids, water, amines, and amino acids, is a limitation of the studies reviewed herein. This omission is due to the technical constraints of the HSI systems employed in these studies, which typically capture data in the VIS range and a less informative portion of the NIR range (700–1100 nm). Additionally, most studies focus on the clinical applicability of HSI within the capabilities of existing diagnostic tools, leading to the exclusion of this spectral range.

Future studies should focus on how difficult it is to automatically analyze data of high spatial and spectral dimension [206]. Another challenge is building a large spectrum database that includes a range of tissue types, from ocular to epidermal and subcutaneous tissue, as well as important molecular biomarkers. A database like this would make it possible to distinguish between different types of tissue, like the bile tube and the surrounding fatty tissue, as well as between oxygenated and deoxygenated blood. The large volume of data generated during HSI affects storage and processing, making HSI too time-consuming to use in particular situations [207]. In addition to this, the cost of HSI equipment can be a barrier to accessibility, restricting its availability to particular users. Future research could focus on validating these techniques on larger, more diverse datasets, which would improve their generalizability. Expanding the datasets to include a wider variety of HNC cases and conditions would provide a better understanding of how these algorithms perform in different clinical scenarios. Moreover, future studies could benefit from multi-center collaborations to collect comprehensive HSI datasets for cross-validation, helping to enhance the robustness and clinical applicability of the CAD techniques.

To overcome these obstacles, a variety of techniques and tools are available. A dataset’s dimensionality can be reduced using dimensionality reduction techniques, which can also be used to manage bands with high correlations [208] in many molecular component blends [209]. The spectral data fusion method is also used for the visualization of HSI image data [210]. Using the spectral scanning method, relatively stable imaging environments might be achieved by progressively capturing spatial images at various wavelengths without requiring the optical system to move mechanically [201]. The snapshot technique has also been developed throughout the years to preserve image speed while enhancing the spectral and spatial resolution [211,212,213], while the immense complexity of HSI data has led to the development of several image analysis techniques, including spectral unmix, statistical analysis, spectral angle mapper, and PCA.

## 5. Conclusions

The studies reviewed herein provide essential insights into the application of HSI and CAD methods for the detection and diagnosis of HNC. Various CAD techniques, including CNNs, SVMs, and LDA, were employed in the studies, and their performance was evaluated across in vivo and ex vivo experiments. LDA demonstrated the most consistent results, with one study achieving 94% accuracy, sensitivity, and specificity in detecting thyroid cancer. In comparison, studies using CNNs showed varying results depending on the cancer type and anatomical site. For instance, the accuracy ranged from 61% for the tongue to 95% for the oropharynx. The sensitivity in specific locations, such as the hypopharynx, was particularly low (20%), despite a high specificity of 99%. This variability indicates that while CNNs perform well for some anatomical sites, further refinement is needed to ensure consistency across all HNC types. SVMs also showed mixed results. One study reported high specificity (89%) but low sensitivity (48%), suggesting that while the SVM effectively reduces false positives, it may miss cancerous lesions, limiting its diagnostic utility in some cases. Overall, in vivo studies outperformed ex vivo experiments, demonstrating higher accuracy (81%), sensitivity (83%), and specificity (79%). This underscores the importance of real-time imaging, where external variables can be more effectively controlled. The studies also revealed that oropharyngeal cancer detection achieved the highest accuracy (95%), while hypopharyngeal cancer detection achieved the lowest (42%). Differences in spectral ranges and resolutions impacted the results, with broader wavelength ranges and higher resolutions contributing to better detection rates. One study utilizing a spectral range of 500–1000 nm and a 2-nm resolution achieved more comprehensive results in tumor margin detection. Despite these promising results, challenges remain, including HSI’s sensitivity to external factors, the need for high-resolution imaging, and the variability in dataset sizes. Future research should focus on larger datasets, more advanced machine learning models, and improved real-time imaging capabilities to make HSI more clinically applicable for HNC diagnosis.

## Figures and Tables

**Figure 1 biomedicines-12-02315-f001:**
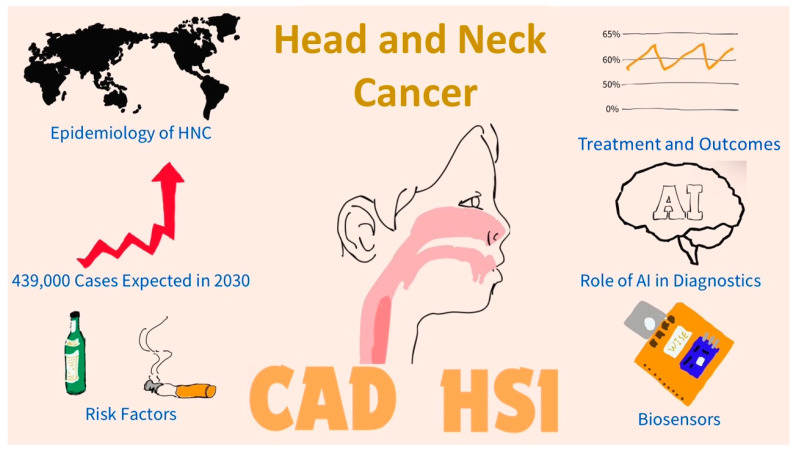
An illustrative diagram of key elements relating to head and neck cancer (HNC), including global incidence and mortality rates, a pie chart of HNC types, affected regions, risk factors, and a timeline of the projected increase in the number of cases by 2030.

**Figure 2 biomedicines-12-02315-f002:**
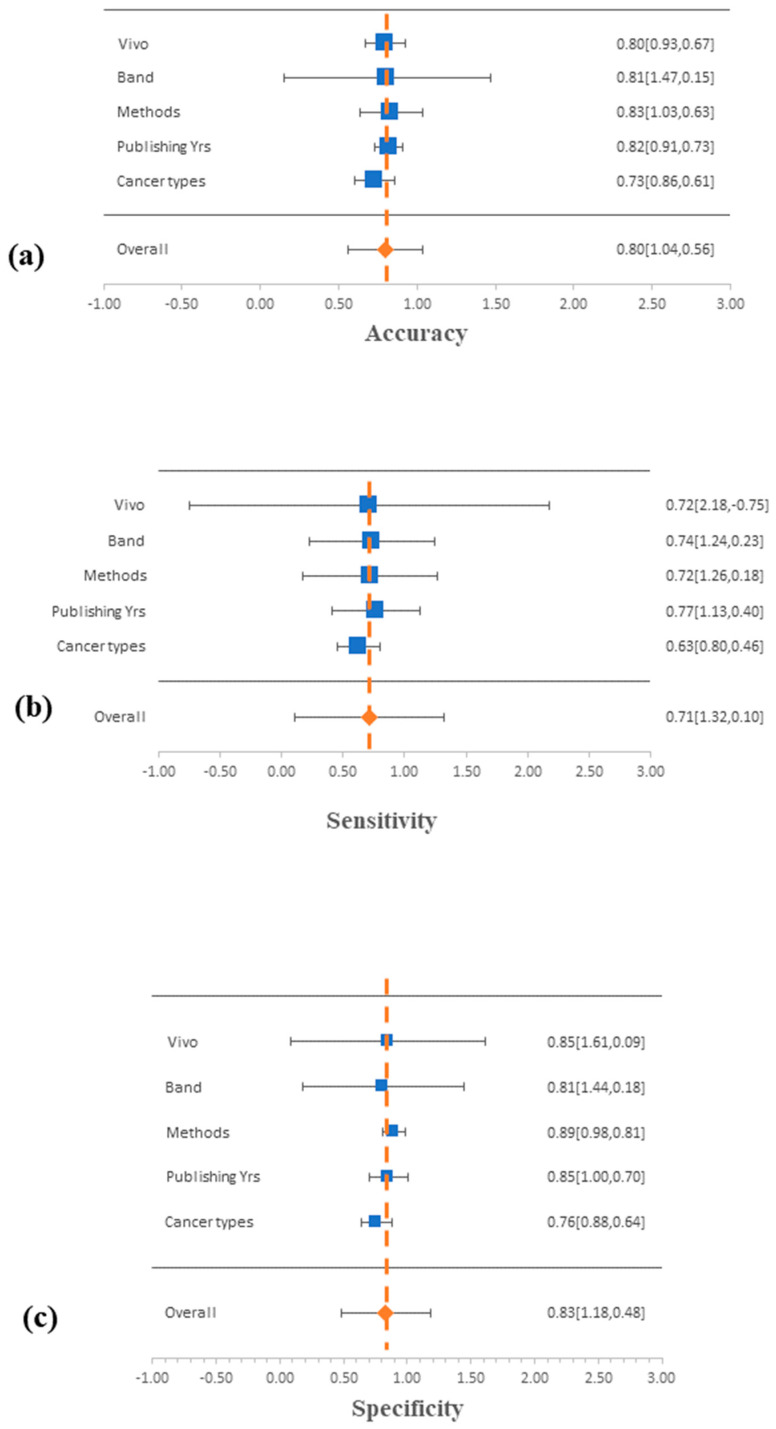
Forest plots of the (**a**) accuracy, (**b**) sensitivity, and (**c**) specificity for different subgroups: cancer type, year of publication, CAD method, spectral band, and experiment type (in/ex vivo).

**Figure 3 biomedicines-12-02315-f003:**
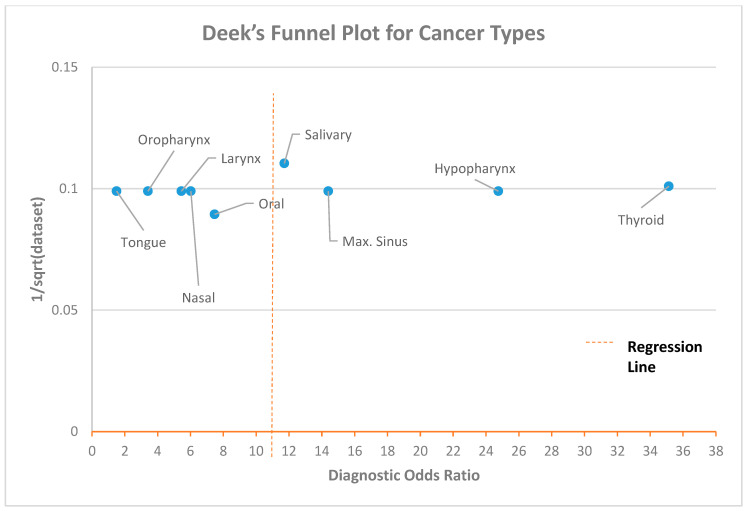
Deek’s funnel plot for cancer types.

**Figure 4 biomedicines-12-02315-f004:**
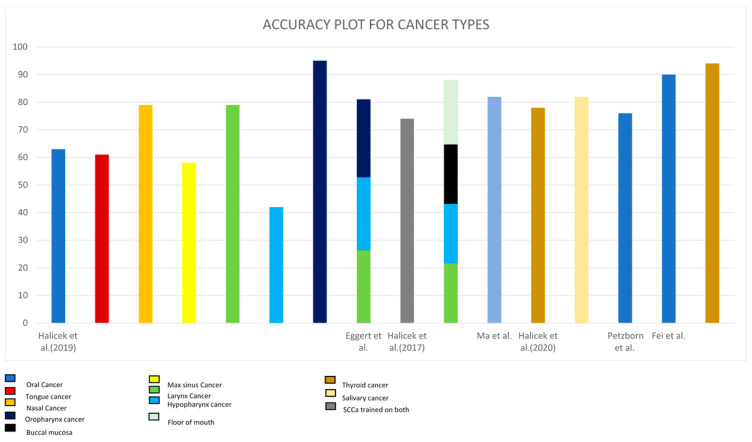
Accuracy plot for cancer types of all the studies chosen in this study Halicek et al. (2019) [166], Eggert et al. (2021) [167], Halicek et al. (2017) [168], Ma et al. (2022) [169], Halicek et al. (2020) [170], Pertzborn et al. (2022) [171], Fei et al. (2017) [172].

**Table 1 biomedicines-12-02315-t001:** Studies on the application of HSI in the medical field, including the detection of various cancers.

Year	Authors	Application	Spectral Range
2023 [143]	Barbosa et al.	Colon cancer	2500–13,333 nm
2013 [144]	Kiyotoki et al.	Gastric cancer	400–800 nm
2020 [145]	Ortega et al.	Breast cancer	400–1000 nm
2020 [146]	Leon et al.	Skin cancer	450–950 nm
2020 [147]	Hsin et al.	Diabetic retinopathy	380–780 nm
2016 [148]	Larisa et al.	Lung cancer	500–670 nm
2012 [149]	Akbari et al.	Prostate cancer	450–950 nm
2021 [150]	Boris et al.	Colorectal cancer	500–1000 nm
2019 [151]	Valentina et al.	Pancreatic cancer	2500–11,111 nm
2018 [152]	Wang et al.	Liver cancer	550–1000 nm
2018 [153]	Fabelo et al.	Brain cancer	400–1000 nm
2021 [154]	Perez et al.	Ovarian cancer	665–975 nm
2022 [155]	Tsai et al.	Esophageal cancer	380–780 nm

**Table 2 biomedicines-12-02315-t002:** QUADAS-2 results for the selected studies.

Study	Risk of Bias	Applicability Concerns
Patient Selection	Index Test	Reference Standard	Flow and Timing	Patient Selection	Index Test	Reference Standard
Halicek et al. [166]		?					
Eggert et al. [167]							
Halicek et al. [168]							
Ma et al. [169]							
Halicek et al. [170]							
Petzborn et al. [171]							
Fei et al. [172]		?				?	


 Low Risk. 
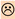
 High Risk. ? Unclear Risk.

**Table 3 biomedicines-12-02315-t003:** Clinical classification of the selected studies.

Authors	Year	Cancer Types	Method	Accuracy	Sensitivity	Specificity	Wavelength	Resolution	In/Ex Vivo	Patients
Halicek et al. [166]	2019	Oral cavity	CNN	63 ± 5%	71 ± 8%	49 ± 8%	450–900 nm	5 nm	In vivo	102
		Tongue		61 ± 7%	57 ± 9%	53 ± 9%				
		Nasal cavity		79 ± 11%	69 ± 17%	73 ± 24%				
		Max. sinus		58 ± 19%	93 ± 5%	52 ± 18%				
		Larynx		79 ± 5%	69 ± 11%	71 ± 9%				
		Hypopharynx		42 ± 9%	20 ± 14%	99 ± 1%				
		Oropharynx		95 ± 4%	49 ± 49%	78 ± 22%				
Eggert et al. [167]	2021	Laryngeal, hypopharyngeal, and oropharyngea	3D-CNN	81%	83%	79%	390–680 nm	10 nm	In vivo	98
Halicek et al. [168]	2017	SCCa trained on both	CNN	74 ± 14%	79 ± 15%	67 ± 20%	450–900 nm	5 nm	Ex vivo	50
		Thyroid trained on both		88 ± 11%	83 ± 23%	92 ± 9%				
Ma et al. [169]	2022	Larynx, hypopharynx, buccal mucosa, and floor of mouth	CNN	82%	72%	93%	470–720 nm	3 nm	Ex vivo	20
Halicek et al. [170]	2020	Thyroid	CNN	78 ± 2%	80 ± 3%	74 ± 3%	450–900 nm	5 nm	Ex vivo	82
		Salivary		82 ± 8%	72 ± 11%	82 ± 11%				
Pertzborn et al. [171]	2022	Oral	SVM	76%	48%	89%	500–1000 nm	2 nm	Ex vivo	7
Fei et al. [172]	2017	Oral cavity	LDA	90 ± 8%	89 ± 9%	91 ± 6%	450–950 nm	2 nm	In vivo	16
		Thyroid		94 ± 6%	94 ± 6%	95 ± 6%				

**Table 4 biomedicines-12-02315-t004:** Meta-analysis and HNC subgroup analysis results for the selected studies.

Subgroup	Number of Studies	Accuracy	Sensitivity	Specificity
Average meta-analysis of all the studies	7	0.77	0.68	0.80
Cancer Type				
Oral	3	0.76	0.48	0.89
Tongue	1	0.61	0.57	0.53
Nasal	1	0.79	0.69	0.73
Max. sinus	1	0.58	0.93	0.52
Larynx	1	0.79	0.69	0.71
Hypopharynx	1	0.42	0.2	0.99
Oropharynx	1	0.95	0.49	0.78
Thyroid	2	0.86	0.87	0.84
Salivary	1	0.82	0.72	0.82
Publishing Year				
2015–2017	2	0.86	0.87	0.86
2018–2021	3	0.81	0.83	0.79
2022–2024	2	0.79	0.6	0.91
Method Used				
CNN	5	0.82	0.77	0.86
SVM	1	0.76	0.48	0.89
LDA	1	0.92	0.91	0.93
In/Ex Vivo				
In vivo	3	0.81	0.83	0.79
Ex vivo	4	0.79	0.6	0.91
Band				
VNIR	5	0.75	0.69	0.76
VIS	2	0.81	0.77	0.86

## Data Availability

The data presented in this study are available upon reasonable request to the corresponding author (H.-C.W.).

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
