# Peer review of "Advancements in Hyperspectral Imaging and Computer-Aided Diagnostic Methods for the Enhanced Detection and Diagnosis of Head and Neck Cancer"

_biomedicines, 2024, doi:10.3390/biomedicines12102315_

Round 1

Reviewer 1 Report

Comments and Suggestions for Authors

The paper examines developments in computer-aided diagnostic (CAD) techniques in conjunction with hyperspectral imaging (HSI) to enhance the identification and diagnosis of head and neck cancer (HNC). The paper examines a number of CAD techniques, including as linear discriminant analysis (LDA), support vector machines (SVM), and convolutional neural networks (CNN). It draws attention to how much better LDA performs and how HSI and CAD have the ability to completely transform HNC diagnoses. Future study directions are suggested as well as challenges including sensitivity to external variables and the requirement for high-resolution imaging are highlighted. Here are a few concerns that are needed to be addressed before the manuscript is good enough for the publication

1. The usage of many CAD techniques, including CNN, SVM, and LDA, is mentioned in the manuscript. Could the authors briefly explain the rationale behind the selection of these specific algorithms for the study and whether any other potentially pertinent algorithms were taken into consideration?

2. The authors point out that HSI is susceptible to outside influences like lighting. If the authors could provide a brief explanation of any methods or procedures they employed to reduce these effects in their trials, that would be very helpful.

3. The publication centers on certain datasets used to verify the techniques. Could the authors provide a quick explanation of how their techniques might be applied to other datasets or provide a brief update on any upcoming plans for validation on a wider range of datasets?

Reviewer 2 Report

Comments and Suggestions for Authors

In this paper, the authors review several carefully chosen papers on the computer-aided diagnostics of head and neck cancer using hyperspectral imaging. This advanced technique determines the spectral characteristics associated with every pixel in the image and, accordingly, contains an enormous amount of information, which has to be thoroughly examined. The authors compare different data analytics approaches to the subject proposed in various publications and make recommendations on their applicability and further development. I think this review can be published in Biomedicines because it draws the attention of researchers working in the specific area of head and neck cancer diagnostics and those who apply the data analytics methods to other types of studies. My minor comment concerns the formatting issues in the abstract, which have to be fixed.

Reviewer 3 Report

Comments and Suggestions for Authors

This paper summarizes the results from 7 studies in the burgeoning and very important field of hyperspectral imaging. No own experiments are presented.  The importance of the paper is to organize the wealth of
of information from the various studies. This is important and interesting for researchers in the field and justifies the publication. In my view, however, some things should be improved:
The number of key points is too small. There are no references e.g. to AI or ML methods

Fig 2 is blurred - improve here if possible
Fig 3 is blurred - dito
Fig 4: Labeling is blurred

Only studies that use data from the UV and VIS range, from an unimportant part of the NIR (700-1100) and the MIR are used. The important part of the NIR, where information about lipids, water, amines and amino acids can be obtained (1000-1900 nm) is excluded. I realize that the integration of this spectral range cannot simply be made up for, but it should be explained in detail why the most important spectral range is excluded.
The discussion is not dialectical enough and lacks any essential message. There are too many
meaningless sentences, such as “ To overcome these obstacles, a variety of techniques and tools are available. Data sets can have their dimensionality reduced using dimensionality reduction techniques, which can also be used to manage bands with high correlation”

The Conclusions are far too short and lacking in content. They contain too many platitudes, such as “The reviewed articles have shown a significant progress in HSI technology”. The authors could write here which lessons can really be learned from the individual studies, perhaps advantages of one or the other study in any applications.

In addition to the many dry numbers, it would also be nice to have some few nice illustrative examples, possibly as eye-catching false-color pictures as a quote from one or the other study.

Round 2

Reviewer 3 Report

Comments and Suggestions for Authors

thank you for integration of the recommended information.

Now, the article should be published.